# Cardioprotective Effects of Honey and Its Constituent: An Evidence-Based Review of Laboratory Studies and Clinical Trials

**DOI:** 10.3390/ijerph17103613

**Published:** 2020-05-21

**Authors:** Ruszymah Bt Hj Idrus, Nur Qisya Afifah Veronica Sainik, Abid Nordin, Aminuddin Bin Saim, Nadiah Sulaiman

**Affiliations:** 1Tissue Engineering Centre, Universiti Kebangsaan Malaysia Medical Centre, Cheras, Kuala Lumpur 56000, Malaysia; ruszyidrus@gmail.com (R.B.H.I.); m.abid.nordin@gmail.com (A.N.); 2Department of Physiology, Faculty of Medicine, Universiti Kebangsaan Malaysia, Cheras, Kuala Lumpur 56000, Malaysia; 2012ronaldvs@gmail.com; 3Ear, Nose & Throat Consultant Clinic, Ampang Puteri Specialist Hospital, Ampang, Selangor 68000, Malaysia; aminuddin_saim@yahoo.com

**Keywords:** honey, myocardial infarction, cardiovascular disease, lipid metabolism, antioxidant

## Abstract

Cardiovascular disease is a major public health burden worldwide. Myocardial infarction is the most common form of cardiovascular disease resulting from low blood supply to the heart. It can lead to further complications such as cardiac arrhythmia, toxic metabolite accumulation, and permanently infarcted areas. Honey is one of the most prized medicinal remedies used since ancient times. There is evidence that indicates honey can function as a cardioprotective agent in cardiovascular diseases. The present review compiles and discusses the available evidence on the effect of honey on cardiovascular diseases. Three electronic databases, namely, PubMed, Scopus, and MEDLINE via EBSCOhost, were searched between January 1959 and March 2020 to identify reports on the cardioprotective effect of honey. Based on the pre-set eligibility criteria, 25 qualified articles were selected and discussed in this review. Honey investigated in the studies included varieties according to their geological origin. Honey protects the heart via lipid metabolism improvement, antioxidative activity, blood pressure modulation, heartbeat restoration, myocardial infarct area reduction, antiaging properties, and cell apoptosis attenuation. This review establishes honey as a potential candidate to be explored further as a natural and dietary alternative to the management of cardiovascular disease.

## 1. Introduction

### 1.1. Burden of Heart Disease

Cardiovascular diseases are estimated to make up 31% of all global deaths [1]. A common presentation of heart disease, myocardial infarction, occurs when the coronary arteries blood supply to the myocardium is low, leading to hypoxia [2]. An infarct could lead to cardiac arrhythmia, one of the common contributors to morbidity in cardiac health [3]. Moreover, ischemia may alter in situ metabolism through inhibition of the lipid metabolism and induction of oxidative stress, leading to the accumulation of their toxic metabolites in the heart [4].

### 1.2. Parameters of Cardiovascular Health

Oxidative stress in heart diseases, especially in ischemia, results in excessive production or accumulation of free radicals or its oxidation products [5]. These free radicals are known as reactive oxygen species (ROS) that originated from oxygen. Endogenously, there is a set of free radical antioxidant enzymes, comprising superoxide dismutase (SOD), catalase, glutathione peroxidase (GPx), glutathione reductase (GRx), and glutathione-S-transferase (GST), that acts as an antioxidant defense system against the ROS [6]. Excess ROS causes oxidative damage to proteins, lipids, and DNA. Ultimately, the damage is manifested in the form of myocardial infarction, following the disruption of the lipid membrane permeability and alteration of the cardiac enzyme activity by the oxidants. Hence, the measurement of ROS and endogenous antioxidant enzymes can be a good predictor of cardiovascular risk [7].

Impaired lipid metabolism can lead to the elevation of total cholesterol, low-density lipoprotein (LDL), and triglyceride (TG). Excessive ROS can attack the elevated LDL to gradually form atherosclerotic plaque [8]. Eventually, the plaque will block the blood supply to the myocardium, inducing hypoxic state that can lead to myocardial tissue necrosis. Having a necrotic tissue in the myocardium can affect the normal beating of the heart. Hence, electrocardiogram parameters such as arrhythmia can be used as a marker of cardiovascular diseases [9].

Cardiac troponin I (cTnI) is a low molecular weight protein constituent of the myofibrillary contractile apparatus of the cardiac muscle [10]. The presence of cTnI in serum shows the release of these proteins to the circulation following an injury to the cardiac muscle. The level of cTnI will increase between 4 to 6 hours after injury and remain elevated up to 5 to 10 days, making it a very specific and sensitive marker for diagnosis of heart attacks [11]. Creatinine kinase MB (CK–MB), aspartate aminotransferase (AST), and alanine aminotransferase (ALT) are some of the enzymes that can be found in the cell of the myocardium [12]. These enzymes, under normal physiological conditions, function in ATP production and cell metabolism intracellularly. Oxygen deprivation in some parts of the heart circulation causes cell death in the myocardium, which can then be clinically diagnosed as a myocardial infarction. Cell deaths in the myocardium, therefore, lead to the release of these biochemical markers that can serve as a predictor of myocardial infarction [13]. Therefore, these markers are included in the general guidelines in the diagnosis management of acute coronary syndrome (ACS) [14,15].

### 1.3. Honey and Its Constituents

Honey is a sweetener that is either consumed by itself or in combination with a variety of foods as an energy source or used for the promotion of health. It is made up of approximately 80% carbohydrate (35% glucose, 40% fructose, and 5% sucrose) and 20% water. About 180 different substances, inclusive of amino acids, vitamins, and minerals, have been reported to be contained in honey [16].

A consensus from a comprehensive review on honey composition revealed that honey contains approximately 1.13% proteins, 0.36% minerals, 215.2 mg/g lipid, 15.5 mg/kg (hydroxymethyl)furfural, 13.2 mg/g vitamin C, 8.57 milliequivalents/kg lactone, and 873.3 mg/kg proline content. Among the major minerals that are known to be detected in honey are sodium, potassium, calcium, magnesium, lead, sulfur, and chloride [16]. 

Manuka honey, the most extensively studied honey, is believed to exert its action via its bioactive constituents. Flavonoids, namely, pinobanksin, chrysin, and pinocembrin, are known to be the bioactive constituents of Manuka honey. Several proven therapeutic potentials of honey include aiding in wound healing [17], antioxidant [18], antimicrobial [19], and anti-inflammatory [20] properties. These proven therapeutic impacts of honey have been associated with its various antioxidant components that exert their effect at the molecular level of disease progression [21].

### 1.4. Cardioprotective Effect of Honey

In terms of honey, several in vitro, in vivo, and clinical trial studies have revealed honey positively affects risk factors for heart problem by improving the plasma lipid profile [22], suppressing oxidation [20], attenuating elevation of cardiac damage markers (CK–MB, AST, ALT) [23], increasing activities of antioxidant enzymes (SOD, GPx, GRx) [24], and increasing LDL resistance to oxidation [25] caused by oxidative stress in heart diseases. In this review, a systematic search of the literature was conducted to identify and discuss all available current evidence that reports the association of honey and cardiovascular diseases to guide the future utilization of honey as a cardioprotective agent.

## 2. Methods 

### 2.1. Literature Search

A search was done systematically in four distinct electronic databases, namely, PubMed, Scopus, Cochrane Library, and MEDLINE via EBSCOhost, to summarize all available evidence on the effects of honey and its constituent on cardiovascular disease. Articles published between January 1959 and April 2020 were obtained using the following two sets of keywords (1) honey* OR pinocembrin* OR pinobanksin* OR chrysin* OR methylglyoxal* AND (2) heart* OR isch* OR cardiac* OR cardiovascular* OR cardioprotective*.

### 2.2. Inclusion/Exclusion Criteria

The electronic database search results were selected according to a pre-set inclusion and exclusion criteria. Inclusion criteria included studies that were published in the English language, having abstracts available, primary literature including any in vitro, in vivo, or clinical trials, and report the association of honey and cardiovascular disease. Exclusion criteria included studies that were published in a language other than English, do not have abstracts available, secondary literature such as review articles, newsletters, letters to the editor, or erratum, and are not related to honey or cardiovascular disease.

### 2.3. Article Selection

Selected articles were screened in three phases before being included in the review. In the first phase, any article title that did not match the inclusion criteria was excluded. In the second phase, abstracts of the remaining articles were screened, and article abstract that did not meet the inclusion criteria were excluded. In the third phase, the full text of the remaining articles was read thoroughly by three independent reviewers to exclude any paper that did not meet inclusion criteria. All reviewers had to agree on the inclusion of selected articles for the review before the data extraction phase began. Any differences in opinions were resolved through discussion between the reviewers.

### 2.4. Data Management

In order to standardize the data collection, all data extraction was performed independently with the use of a data extraction form. The following data were recorded from the studies: (1) the experimental model or study population used; (2) type or form of honey or its constituent used; (3) the outcome measures; (4) a brief description of the results of the study; (5) the conclusion of the study.

### 2.5. Risk of Bias Assessment

The three independent reviewers evaluated the risk of bias of the included studies using an adapted version of the Office of Health Assessment and Translation (OHAT) risk of bias tool for in vivo studies [26]. The tool is also applicable to access the potential risk of bias of in vitro studies. This tool of assessment includes the risk of bias in the following domains: (1) selection bias; (2) performance bias; (3) detection bias; (4) attrition bias; (5) reporting bias. Studies were judged as having a low risk of bias (+), high risk of bias (-), unclear risk of bias (?), and not applicable (NA). For the clinical studies, Cochrane Collaboration’s tool for assessing the risk of bias was used [27]. Any disagreement on the risk of bias assessment was resolved by further discussion between the reviewers.

## 3. Results

### 3.1. Study Selection

The literature search identified 21,250 potentially relevant records. Initial screening of records resulted in the removal of 13,655 duplicate records before another 446 records that were not original articles or not published in the English language were excluded. Three reviewers then independently assessed all articles based on the title for any exclusion criteria, which resulted in the removal of 10,835 articles. From the remaining 35 articles, 7 articles were removed after screening the abstracts for the inclusion and exclusion criteria. From the remaining 26 articles that fulfilled the inclusion and exclusion criteria, one article [28] was found to be a duplicate study of [29] and was excluded. A total of 27 articles were retrieved for further assessment and data extraction to be included in this review. Differences in opinion between the reviewers regarding the inclusion or exclusion of the full articles were resolved by discussion. A flow chart of the selection process, including reasons for exclusion, is shown in Figure 1.

### 3.2. Study Characteristics

All studies included in the review were published between the years 2002 and 2020. In terms of honey used, it varied from one study to another. In terms of country of origin, six studies obtained their honey from Iran [29,30,31,32,33,34], three from Malaysia [35,36,37], two from Nigeria [22,38], and one each from Bangladesh [23], United Arab Emirates [39], Egypt [24], France [25], Indonesia [40], Turkey [41], and Yemen [42]. One study compared honey from different countries, namely, America, Australia, Germany, and Pakistan [20]. Floral origin of honey was only mentioned in the three studies from Malaysia. Two studies used Tualang honey [35,37], and one study used Gelam honey [36]. In terms of honey constituents, two active compounds of honey were identified, namely, chrysin and pinocembrin. Five studies used chrysin [43,44,45,46,47], and three studies used pinocembrin [48,49,50] as their subject matter.

With regard to the pre-clinical experimental studies, five in vitro studies and seventeen in vivo studies were retrieved. Four in vitro studies were done by introducing experimental ischemia and reperfusion cycle to the excised rat’s heart tissue [29,30,31,32]. Damage to the vascular endothelial cell lining can initiate a cascade of excessive oxidative bursts mediated by resident phagocytes that are very relevant to the myocardial infarction etiology. Hence, one in vitro study that looks into the potential of honey in preventing bovine thrombin-induced oxidative bursts in phagocytes was also included in this review [20]. 

Within the in vivo studies, the effect of honey on cardiovascular parameters was assessed in healthy rats and diseased rat models. In healthy rats, two studies investigated the antihyperlipidemic effect of honey consumption [22,38], and one study each that investigated its effect on bleeding time [40] and cardiac muscle tissue lipids [41]. In diseased rat models, a myocardial injury was introduced to rats via artery ligation in four studies [43,44,48,49], isoproterenol (ISO) injection in three studies [23,37,45], and epinephrine injection, [24] chronic hypoxia [46], Nω-nitro-L-arginine methyl ester (L-NAME) [47], and ischemia and reperfusion cycle [50] in one study each. Other cardiovascular risk factors, such as hypertriglyceridemia [25] and aging [36], were also studied in fructose-induced rats and aged rats, respectively. The summary of the characteristics of all pre-clinical studies is displayed in Table 1.

In humans, four randomized controlled trials and one non-randomized trials were retrieved. All of the human trials sought to elucidate the effect of honey on the modulating parameters associated with cardiovascular risk factors, such as body mass index (BMI), C-reactive protein elevation, hypertension, and abnormal lipid profile [33,34,35,39,42]. The summary of the characteristics of all pre-clinical studies is displayed in Table 2.

### 3.3. Clinical outcome of Honey Consumption

In humans, Al-Waili (2004) conducted a series of clinical studies that investigated the effect of honey on a number of metabolic parameters, including cardiovascular risk markers such as C-reactive protein (CRP), homocysteine, and hyperlipidemia [39]. In both healthy subjects and subjects with diabetes mellitus, honey decreased the levels of plasma glucose, insulin, CRP, total cholesterol, LDL-C, and TG, together with the elevation of HDL-C [39]. 

Honey’s positive effect in the patient’s lipid profile was also reported in a randomized trial done by Yaghoobi et al. (2008) [33]. They reported that honey caused a reduction of serum triacylglycerol, C-reactive protein (CRP), total cholesterol, and LDL-C in their obese participants [33]. In a more recent study done on young, healthy patients, the consumption of honey resulted in the reduction of total cholesterol and LDL, while increasing triglycerol and HDL [34]. 

In terms of blood pressure parameters, a parallel two-arm study using Tualang honey and its mixture with other bee products (95% honey, 4% bee bread, 1% royal jelly), demonstrated a reduction in diastolic blood pressure when consumed for 12 months. The reduction was more statistically significant in the Tualang honey group as opposed to the honey cocktail group [35]. Together, these findings support the positive effect of honey on modulating hypertension.

In pediatric patients, Abdulrhman et al. (2017) investigated the effect of Ziziphus honey from Yaman on the ejection fraction and fraction shortening outcomes of patients with idiopathic dilated cardiomyopathy. The patients were allocated into two groups, both receiving a standard therapy of furosemide, spironolactone, and captopril, each in a dose of 2 mg/kg daily. Only the experimental group received additional oral administration of 1.2 g/kg of honey daily for 3 months. At the end of the study, both groups demonstrated increased ejection fraction and fraction shortening compared to baseline. The honey group demonstrated a significantly higher rate of increase of both parameters when compared to the control group [42].

### 3.4. Honey Improves Lipid Metabolism

A high fructose diet has been associated with a higher incidence of cardiovascular diseases. The fact that fructose makes up approximately 30% of whole honey weight has prompted Busserolles et al. (2002) to investigate the effect of honey on fructose-induced hypertriglyceridemia [25]. As expected, fructose-fed rats demonstrate higher triglyceride (TG) levels. Interestingly, despite its known fructose content, honey-fed rats showed lower TG levels. A lower level of lipid peroxidation was also observed in the heart tissue of honey fed rats compared to the fructose-fed rats [25].

Rhododendron honey, notoriously known as mad honey, has been known to be poisonous in humans and animals due to the toxic compound grayanotoxin. However, in traditional medicine, it is used for the treatment of a variety of diseases such as hypertension and sexual dysfunction [41]. In the cardiac muscle tissue, administration of 50 mg/kg but not 25 mg/kg Rhododendron honey caused significant changes in the lipid molecules [41].

In healthy rats, levels of HDL, TG, CVPI, and VLDL were significantly higher in honey-fed rats compared to control rats. Conversely, the total cholesterol was significantly lower in honey-fed rats compared to control rats [22]. Honey also has an effect on the post-prandial lipid metabolism in rats. Pre-feeding of honey before a high-fat diet resulted in significantly higher post-prandial HDL, cholesterol, and HMG-CoA:mevalonate, while it significantly lowered post-prandial LDL and cholesterol [38]. In a myocardial injury rat model, pretreatment with Sundarban honey significantly reduced the level of serum TC, TGs, and LDL-C, while significantly increased the level of HDL-C when compared to the control group [23].

### 3.5. Honey Antioxidative Effects

In fructose-induced hypertriglyceridemia, the elevation of nitric oxide and reduction of α-tocopherol indicate an increase in oxidative stress in rats. Interestingly, despite its known fructose content, honey-fed rats have lower nitric oxide and elevation of α-tocopherol [25]. In another study, the supplementation of honey in ISO-induced myocardial injury restored the activity of GRx and improved in the scavenging activities of SOD and GPx that was impaired with ISO induction [23]. Honey treatment was also shown to restore several perturbations induced by epinephrine, namely, GSH, ascorbic acid (vitamin C), and SOD levels [24].

### 3.6. Honey Prevents Free Radical Production

Honey effects on suppressing the bovine thrombin-induced phagocytic oxidation could be extremely beneficial in the interruption of the pathological progress of cardiovascular disease. Ahmad et al. (2009) evaluated the production of ROS by neutrophil and macrophage activated with bovine thrombin in the presence of honey [20]. When compared to the control, the presence of honey was able to completely suppress the production of ROS in both thrombin-induced neutrophils and macrophages [20].

### 3.7. Honey Modulates Blood Pressure

In epinephrine-induced cardiac injury rats, venous blood pressure (VBP) was elevated in response to epinephrine. Pretreatment with honey reverses this elevation, indicating modulation of honey on vasomotor function [24]. In another study that utilized bleeding time to study the effect of honey on modulating blood pressure, honey resulted in slower bleeding time as compared to the control [40]. Together, these findings support the positive effect of honey on modulating hypertension.

### 3.8. Honey Ameliorates Cardiac Arrhythmia

Development of the ex vivo cardiac arrhythmia model by Najafi et al. in 2011 enabled studies on the cardioprotective effect of honey to go beyond lipid profiles and free radicals. The model was established in isolated rat heart that was induced by ischemia and reperfusion cycle [30]. Reperfusion with honey (1% and 2%) following ischemia showed a reduction in the number of ventricular tachycardia (VT), ventricular ectopic beats (VEBs), and the incidence of VT and total ventricular fibrillation (VF). Moreover, honey (2%) was found to significantly reduce the incidence and duration of reversible VF [30]. 

The same group continued to explore the effect of different concentrations of honey on their ischemia/reperfusion model. In 2012, they reported that honey at 0.25%, 0.5%, 1%, and 2% [24], as well as 0.125%, 0.25%, 0.5%, and 1% [29], and showed a decrease in VEB number, a reduction in number and duration of VT compared to control. In 2013, the administration of 1%, 2%, and 4% honey showed a reduction in the number of VT and VEBs at 30 min in ischemic conditions [31].

Electrocardiogram (ECG) data of abnormalities, such as extrasystole, tachyarrhythmia, bradyarrhythmia, and abnormalities of both P-wave and ST segments, have also been reported in the earlier epinephrine-induced cardiac disorder rat models. Pretreatment with honey has been shown to restore the heart back to normal [24].

### 3.9. Honey Reduces Myocardial Infarct Area

Another advantage of the ex vivo model of ischemia and reperfusion cycle in isolated rat heart is the ability to measure myocardial infarction area size. The Najafi group reported a significant reduction in infarct size with different degrees, depending on the concentration of the honey in comparison to the untreated control [29,31,32]. In an ISO-induced myocardial injury rat model, Khalil et al. also evaluated the infarction area via histopathological analysis. Treatment of honey was found to decrease the degree of infiltration of inflammatory cells and has relatively well-preserved cardiac muscle fiber morphology, indicating a protective effect of honey against myocardial infarction [37].

### 3.10. Honey Reduces Myocardial Injury Marker

The cardioprotective effect of honey was corroborated with the data of myocardial injury marker in ISO-induced myocardial injury in rats. With ISO-induction elevation of cTn-I, CK-MB, LDH, AST, and ALT were apparent. Pretreatment with Sundarban honey [23] and Tualang honey [37] has shown to significantly decrease the level of serum cTn-I, CK-MB, LDH, AST, and ALT.

### 3.11. Honey Modulates Age-Related Protein Expression

One of the known risk factors for cardiovascular disease is old age. Aging has been associated with the decline of defense and repair mechanism of the body system, which leads to physiological dysfunction, homeostasis imbalance, and, consequently, cell death. Hasenan et al. (2018) investigated the differential protein expression between young (2 months old) and old (19 months old) rats [36]. When comparing between the young and old rats, 69 proteins were differentially expressed. Out of these, 68 of them were downregulated, and only 1 protein was upregulated. In old rats, the treatment of honey revealed 55 proteins were differentially expressed. Out of these, 52 of them were upregulated, and only 3 proteins were downregulated. Analysis of the data revealed that supplementation of honey was found to upregulate most of the downregulated protein in aged rats, and these proteins play a substantial role in the electron transport chain biochemical pathway [36].

### 3.12. Chrysin Rescues Myocardial Injury

Three studies included in this review focus on a known honey bioactive compound, chrysin, and its effect on myocardial injury rescue. In the first two studies, the left anterior descending coronary artery of the rat was ligated to induced acute myocardial infarction. In the third study, diabetic rat induced with streptozotocin was injected with ISO to produce a myocardial injury in a diabetes rat model. Treatment of chrysin successfully reversed the diminished hemodynamic and ventricular functions observed in ISO-induced rats. In terms of antioxidant activities, the treatment of chrysin significantly reduced the oxidative stress caused by ISO-induction, artery ligation, and PPAR-γ inhibition. Treatment with chrysin also reverses PPAR-γ inhibition and significantly suppressed RAGE, NF-κBp65, and IKK-β protein expressions and TNF-α level. Cardiac injury markers were also significantly reduced with the treatment of chrysin. Treatment of chrysin shows cardiovascular protection via the reduction in apoptosis marker. Histology observation confirms the preservation effect that chrysin has on the myocardium structure [43,44,45,46,47].

### 3.13. Chrysin Regulates Blood Pressure

Regarding chrysin, two out of the five studies also explored the role of chrysin in regulating blood pressure. In hypoxia-induced hypertensive rats, ventricular blood pressure was elevated, as indicated by the increased right ventricular systolic pressure (RVSP) and mean right ventricular pressure (mRVP). Pretreatment with honey reverses this elevation, indicating modulation of chrysin on pulmonary hypertension [46]. In another study that utilized Nω-nitro-L-arginine methyl ester (L-NAME) to induce hypertension, chrysin reversed the elevated left ventricular functions and angiotensin-II, as well as the suppression of the cardiac HO-1 and cGMP levels [47]. Together, these findings support the positive effect of chrysin on modulating hypertension.

### 3.14. Pinocembrin Ameliorates Cardiac Arrhythmia

Three studies included in this review focus on another bioactive compound of honey, pinocembrin. In the first and second studies, the left anterior descending coronary artery of rat was ligated to induced acute myocardial infarction. In the third study, the ischemia and reperfusion cycle model of myocardial infarction was used. In the atrial ligation model, pinocembrin treatment significantly ameliorates cardiac arrhythmia as indicated by improved heart rate variability, shortened atrial activation latency, and prolonged atrial effective refractory period [48,49]. In parallel, the reperfusion model also demonstrated the positive effect of pinocembrin when pinocembrin increased heart rate, mean arterial pressure, and rate–pressure product parameters in echocardiograph [50].

### 3.15. Risk of Bias Assessment

In general, the in vitro and in vivo studies included have a low risk of bias. All studies have an equal allocation of the exposed and unexposed group, preventing selection bias towards the conclusion. Experimental conditions were kept constant across all studies, preventing confounding elements from affecting the outcome of the studies. In animal studies, blinding of the study group during the course of the study is often not possible for animal welfare considerations and the need to determine if treated animals are affected relative to controls in a treatment- or dose-dependent manner. Hence, many of the included studies have a high risk of bias in outcome assessment. All studies have a low risk of attrition, detection, and reporting bias. 

For clinical studies, only Al Waili et al. (2004) were graded as having a moderate risk of bias. This is because the study comprises a series of 6 distinct mini studies. As a result, some of the treatment did not have a comparison group available. The remaining four studies demonstrated a low risk of bias in terms of selection, performance, detection, attrition, and reporting bias due to their randomized controlled trial design. 

The results of the risk assessment of the pre-clinical studies and clinical studies are summarized in Table 3 and Table 4, respectively.

## 4. Discussion

Throughout history, many drugs have been developed for the management of cardiovascular diseases. However, many of the drugs come with adverse effects. Hence, finding a dietary supplement with cardioprotective properties has been an emerging trend among medical scientists.

The Lyon Diet Heart Study is a randomized controlled trial that investigated the cardioprotective effect of the Mediterranean diet. The outcome analysis from this study revealed a striking cardioprotective effect of the Mediterranean diet after 27 months of follow-up [51]. 

The interest of using honey as a cardioprotective agent started with the discovery of its antilipidemic effect in a fructose-induced hypertriglyceridemia rat model [25]. The study also revealed antioxidative effects and protection from lipid peroxidation by honey. Following that, several other pre-clinical and clinical studies have confirmed the beneficial effect of honey in lowering the risk factor for cardiovascular diseases [22,23,34,39].

Many diseases have been associated with the imbalance between oxidative stress via the production of ROS and its suppression by endogenous antioxidant enzymes. This includes cardiovascular diseases. The antioxidant properties of honey have been reported extensively [18,19,21]. The antioxidant capacity differs greatly depending on the honey floral source. Variance in the phenolic content of honey from different sources has been attributed to their antioxidant activity [52,53]. Honey inhibition of ROS production has also been reported. Similarly, the degree of inhibition also differs according to their floral origin [54].

In this review, honey from different origins were shown to elevate the activity of antioxidant enzymes in myocardial injury rat models [23,24,25]. In terms of ROS inhibition, one study using neutrophil and macrophage activated with bovine thrombin reported complete inhibition of ROS production when honey was present [20].

With the development of the myocardial injury model in animals, more parameters relevant to cardiovascular diseases can be measured. For example, cardiac arrhythmia and myocardial infarct size can be measured from the ex vivo model created by subjecting isolated rat heart to ischemia and reperfusion cycle. Honey has been proven repeatedly to restore heartbeat and reduce myocardial infarct size [29,30,31,32].

Aging can also be a contributing risk factor to cardiovascular diseases. A mass spectrometry investigation comparing young and old rats revealed differential protein expression induced by aging. Treatment with honey appears to revert the aging-related protein expression change. This result suggests the antiaging effect of honey against age-related pathology, including cardiovascular diseases [36]. Honey also provides protection at the cellular level when chrysin, a bioactive compound found in honey, was proven to attenuate apoptosis and restore the PPAR-γ activity in rat’s heart [36].

## 5. Conclusions

The review has successfully identified relevant evidence on the honey cardioprotective effect. Honey is able to modulate oxidation, reduce blood pressure, restore heartbeats, reduce myocardial infarct areas, improve lipid metabolism, exert antiaging properties, and attenuate cell apoptosis. Honey is demonstrated to be a potential candidate as a natural alternative for the management of cardiovascular disease.

## Figures and Tables

**Figure 1 ijerph-17-03613-f001:**
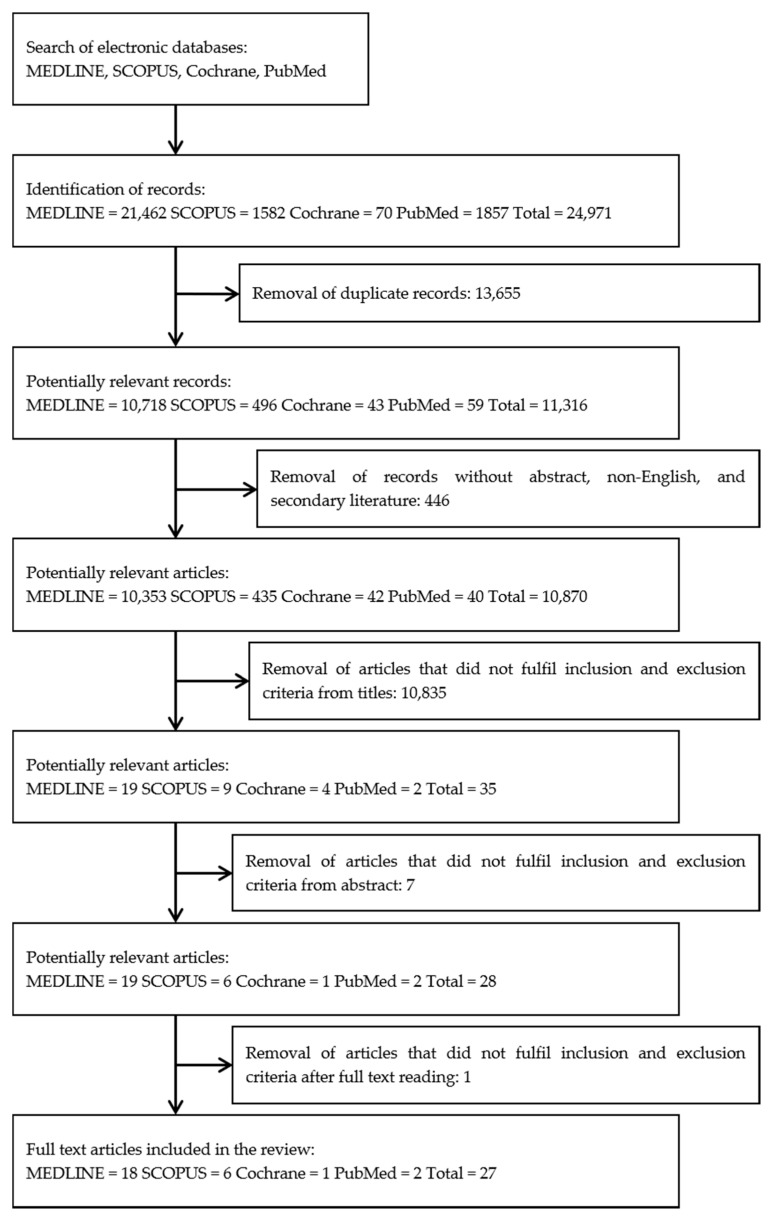
Flow chart of the selection procedure to identify relevant reports on honey’s cardioprotective effect.

**Table 1 ijerph-17-03613-t001:** Pre-clinical evidence on the cardioprotective effect of honey.

**Honey improves Lipid Metabolism**					
**Ref**	**Experimental Model**	**Honey Treatment**	**Outcome Measures**	**Result**	**Conclusion**
[41]	Normal male mice (*n* = 18)	Intragastric delivery of 25 or 50 mg/kg Rhododendron honey 24 h before analysis.	ATR–FTIR spectroscopy of cardiac muscle tissue lipids.	Administration of 50 mg/kg but not 25 mg/kg honey caused significant changes in the lipid molecules.	Consumption of honey causes significant toxic effects on cardiac muscle tissue lipids.
[38]	Wistar rats fed with high-fat diet or portable water (*n* = 25)	Oral delivery of 1 g/kg Nigerian honey 5 or 60 min prior to administration of high-fat diet.	TC TG LDL Very low-density lipoprotein (VLDL) Cholesterol Catalase HDL 3-hydroxy-3-methyl-glutaryl-coenzyme A (HMG-CoA)	Honey pretreatment resulted in significantly higher post-prandial HDL, cholesterol, and HMG-CoA: mevalonate while significantly lowered post-prandial LDL and cholesterol.	Pretreatment of honey has post-prandial antihyperlipidemic effect.
[22]	Normal adult male albino rats (*n* = 20)	10% (*v/v*) Nigerian honey (Nigeria) added to the rat drinking water for 22 weeks.	TG TC LDL Very low-density lipoprotein (VLDL) Cardiovascular risk predictive index (CVPI) Plasma cholesterol HDL	Treatment of honey reduced the level of VLDL, LDL, TG, CVPI, plasma cholesterol, TC, and HDL.	Honey administration reduced risk of coronary heart disease in male albino rats.
[23]	Isoproterenol (ISO)-induced myocardial injury model in adult Wistar rats (*n* = 72)	5 g/kg oral Sundarban honey (Bangladesh) for 6 weeks.	Serum cardiac troponin 1 (cTn-1) Cardiac marker enzymes Serum lipid profile Antioxidant activities	Treatment of honey restores the ISO-induced elevation of cardiac troponin, cardiac enzymes, serum lipid profile, and lipid peroxidation.	Sundarban honey confers protection against ISO-induced oxidative stress in the myocardium.
**Honey antioxidative effects**					
**Ref**	**Experimental Model**	**Honey Treatment**	**Outcome Measures**	**Result**	**Conclusion**
[25]	Fructose-induced hypertriglyceridemic and oxidation in Weaning male Wistar rats (*n* = 27).	65 g/100 g oral Ceyrat honey (France) in normal rat feed for 2 weeks.	TG α-tocopherol level Plasma nitrite and nitrate Lipid peroxidation Thiobarbituric acid-reactive substances (TBARS).	Treatment of honey lead to decreased TG, higher α-tocopherol level, lower plasma nitrite and nitrate concentrations, lower lipid peroxidation, and lower TBARS.	Substituting honey for refined carbohydrates protects from hypertriglyceridemia and prooxidative effects of Fructose in nutritional diet.
[20]	Human neutrophils or rodent peritoneal macrophages stimulated with various concentrations of bovine thrombin (0.5 to 0.0002 units/mL).	0.03 to 15 mg/ml of six honey samples: Clover honey (America), Capilano honey (Australia), Langnease honey (Germany), Al-Shafa honey (Pakistan), Swat honey (Pakistan), Sidder honey (Pakistan).	Oxidative activity determined by oxidation of luminol or lucigenin.	Treatment of 1 mg/mL honey completely suppressed oxidative activities.	Natural honey suppression of oxidation could be extremely beneficial by attenuating the progress of cardiovascular disease.
**Honey modulates blood pressure**					
**Ref**	**Experimental Model**	**Honey Treatment**	**Outcome Measures**	**Result**	**Conclusion**
[40]	Double Ditsch Webster mice (*n* = 32)	High Desert® 75 mg honey oral for 12 days.	Bleeding time	Treatment of honey resulted in slower bleeding time as compared to the control.	Honey has an antiplatelet effect in mice.
[24]	Epinephrine-induced cardiac and vasomotor dysfunction in adult male Sprague–Dawley albino rats (*n* = 30).	5 g/kg intraperitoneal honey (Saint Katherine Protectorate, Southern Sinai, Egypt).	Venous blood pressure (VBP) ECG parameters Total antioxidant capacity	Treatment of honey reverses epinephrine-induced elevation of VBP, abnormalities in extrasystoles, tachyarrhythmias, bradyarrhythmia, and decreased in total antioxidant capacity.	Honey has a cardioactive and vasoactive effect that can play a crucial role as a cardioprotective agent.
**Honey ameliorates cardiac arrhythmia**					
**Ref**	**Experimental Model**	**Honey Treatment**	**Outcome Measures**	**Result**	**Conclusion**
[30]	Ischemia and reperfusion cycle myocardial injury model in male Wistar rats (*n* = 50–70).	1, 2, and 4% (*v/v*) honey (Oskou, East Azerbaijan, Iran) added in rats drinking water for 45 days.	Ventricular tachycardia (VT) Ventricular ectopic beats (VEBs) Ventricular fibrillation (VF)	Treatment of honey reduced the number of VT, VEBs and incidence of VT, total VF incidence and duration of reversible VF following reperfusion cycle.	Long-term oral administration of honey can recover ischemic-reperfused isolated rat hearts and consequently has anti-arrhythmic activity.
[31]	Ischemia and reperfusion cycle myocardial injury model in male Wistar rats (*n* = 40–56).	1, 2, and 3% (*v/v*) honey (Oskou, East Azerbaijan, Iran) added in rats drinking water for 45 days.	Myocardial infarct size Ventricular tachycardia (VT) Ventricular ectopic beats (VEBs) Ventricular fibrillation (VF)	Treatment of honey reduced infarct size, VT, VEB, incidence of VT and total VF following reperfusion cycle.	Honey showed cardioprotective effects in in vivo long-term pretreatment following myocardial infarction.
**Honey reduces myocardial infarct area**					
**Ref**	**Experimental Model**	**Honey Treatment**	**Outcome Measures**	**Result**	**Conclusion**
[29]	Ischemia and reperfusion cycle myocardial injury model in male Wistar rats (*n* = 50–70).	0.25%, 0.5%, 1%, and 2% (*v/v*) honey (Oskou, East Azerbaijan, Iran) added in rats drinking water for 45 days.	Myocardial infarct size Ventricular tachycardia (VT) Ventricular ectopic beats (VEBs)	Treatment of honey reduced the number of single ectopic and reduced number of arrhythmias, the total number of VEBs and duration of VT, and percentage of infarct size and infarcted volume following reperfusion cycle.	Post-ischemic administration of natural honey in global ischemia showed protective effects against ischemia/reperfusion (I/R) injuries in isolated rat heart.
[32]	Ischemia and reperfusion cycle myocardial injury model in male Sprague-Dawley rats (*n* = 32–40).	0.125%, 0.25%, 0.5%, and 1% (*v/v*) honey (Oskou, East Azerbaijan, Iran) added in rats drinking water for 45 days.	Arrhythmia Ventricular tachycardia (VT) Ventricular ectopic beats (VEBs) Ventricular fibrillation (VF)	Treatment of honey reduced the number, the duration and the incidence of recorded arrhythmias, the duration and the incidence of VT and reversible VF, number of VEBs and VT and the time spent in reversible VF and VT, the duration and the incidence of reversible VF and total VF following reperfusion cycle.	The long-term administration of natural honey also caused significant cardioprotection against the myocardial infarction.
[37]	ISO-induced myocardial injury model in adult Wistar albino rats (*n* = 40)	3 g/kg oral Tualang honey (Malaysia) for 45 days.	Heart weight Cardiac troponin Cardiac enzymes TC TG Antioxidant activity	Treatment of honey reverses ISO-induced heart enlargement, elevation of cardiac troponin, cardiac enzymes, serum total cholesterol, and triglyceride, as well as decreased antioxidant activity.	Cardioprotection of Tualang honey against cardiovascular diseases.
**Honey modulates age-related protein expression**					
**Ref**	**Experimental Model**	**Honey Treatment**	**Outcome Measures**	**Result**	**Conclusion**
[36]	Young (2 months) and old (19 months) Sprague–Dawley male rats (*n* = 12).	2.5 g/kg oral Gelam honey (Malaysia) for 8 months.	Differential expression of mitochondrial protein.	Treatment of honey restores the decreased expression of protein related to oxidative phosphorylation caused by aging such as ATP synthase, NADH dehydrogenase, and superoxide dismutase.	Gelam honey provides protective effect on cardiac tissue of aged rats by modulating age-related protein expressions.
**Chrysin improves myocardial injury**					
**Ref**	**Experimental Model**	**Honey Treatment**	**Outcome Measures**	**Result**	**Conclusion**
[43]	The left coronary artery of adult male Sprague-Dawley rats was ligated to induced acute myocardial infarction. (*n* = 30)	50 mg/kg intragastric chrysin daily for 5 days.	Myocardial infarct size. Myocardial histology Electrocardiographs Inflammatory cytokines (TNF-α, IL-6, and IL-1β) Cardiac marker enzymes	Chrysin reduced artery ligation-induced infarct size, inflammatory cells in the myocardial tissue, elevation of inflammatory cytokines and elevation of cardiac marker enzymes.	Chrysin attenuates myocardial injury by inhibiting myocardial inflammation.
[44]	The left anterior descending coronary artery of adult male Sprague–Dawley rats was ligated to induced acute myocardial infarction.	40 mg/kg oral for4 weeks.	Echocardiograph Masson trichrome staining Immunochemistry Western blot RT-qPCR	Chrysin improves cardiac systolic function, alleviates oxidative stress, alleviates interstitial and perivascular fibrosis, reduces the expression of type I collagen, reduces the NF-_K_B p65 level and p-IKKβ/IKKβ ratio, reduces the expression levels of c-Fos and c-Jun, reduces ANGII-induced up-regulation of type I and type III collagen levels	Chrysin inhibits myofibroblast transformation and collagen synthesis, prevents myocardial fibrosis, and improves cardiac function.
[45]	Streptozotocin-induced diabetic male albino Wistar rats challenged with ISO to induce myocardial injury (*n* = 75).	60 mg/kg oral chrysin for 28 days.	Ventricular functions Antioxidant activities Cardiac injury markers Apoptosis markers Myocardium structure	Treatment with chrysin restores the ISO-induced ventricular and myocardium damage, elevation of oxidative stress, cardiac injury markers, and apoptosis effect.	Chrysin ameliorated ISO-induced myocardial injury in diabetic rats through PPAR-γ activation.
**Chrysin regulates blood pressure**					
**Ref**	**Experimental Model/Study Population**	**Honey Treatment**	**Outcome Measures**	**Result**	**Conclusion**
[46]	Pulmonary hypertension was established in Sprague–Dawley rats via exposure to chronic hypoxia (*n* = 20)	100-mg/kg chrysin injected via subcutaneous daily for 21 days.	Right ventricular systolic pressure (RVSP) Mean right ventricular pressure (mRVP) Right ventricular hypertrophy Gene and protein expression of canonical transient receptor potential channel (TRPC), hypoxia inducible factor α (HIF-α), and bone morphogenetic protein (BMP).	Chrysin resulted in reversal of the hypoxia-induced RVSP and mRVP decline while attenuating right ventricular hypertrophy and increasing levels of hypoxia-related genes and proteins.	Chrysin display cardioprotective effect in hypoxia-induced pulmonary hypertension.
[47]	N^ω^-nitro-L-arginine methyl ester (L-NAME)-induced hypertension in male Wistar rats (*n* = 24)	25 mg/kg oral for4 weeks.	Cardiac and vascular function Angiotensin II (Ang-II) levels Hexo oxygenase (HO-1) levels Cyclic guanosine monophosphate (cGMP) levels.	Chrysin treatment reverses the L-NAME-induced elevated left ventricular functions and Ang-II while increases the cardiac HO-1 and cGMP levels.	Chrysin exerts antihypertensive effects via angiotensin system.
**Pinocembrin ameliorates cardiac arrhythmia**					
**Ref**	**Experimental Model**	**Honey Treatment**	**Outcome Measures**	**Result**	**Conclusion**
[48]	The left anterior descending coronary artery of adult male Sprague–Dawley rats was ligated to induced acute myocardial infarction (*n* = 106)	5 mg/kg pinocembrin intravenous daily for 6 days before injection of *E. coli*.	Heart rate variability (HRV). Atrial activation latency (AL). Effective refractory period (ERP). Degree of fibrosis. Norepinephrine (NE), TNF-α, IL-1β, and IL-6 levels. Expression of Cx43 and Cav1.2 Phosphorylation of Iκβα and p65.	Pinocembrin treatment significantly improved HRV, shortened atrial AL, prolonged atrial ERP, attenuated atrial fibrosis, and decreased concentrations of norepinephrine (NE), tumor necrosis factor-α (TNF-α), interleukin (IL)-1β and IL-6, increased expression of Cx43 and Cav1.2 and suppressed the phosphorylation of IκBα and p65.	Pinocembrin is protective against atrial arrhythmia.
[49]	The left anterior descending coronary artery of adult male Sprague–Dawley rats was ligated to induced acute myocardial infarction (*n* = 45)	30 mg/kg pinocembrin intravenous 10 min before ligation.	HR MAP RPP Arrhythmia index. Na^+^-K^+^-ATPase and Ca^2+^-Mg^2+^-ATPase activities CK-MB and cTnI levels. Gene and protein expression of Cx43 and Kir2.1.	Pinocembrin alleviates the surgical-induced reduction of HR, MAP, RPP, Na^+^-K^+^-ATPase activities, Ca^2+^-Mg^2+^-ATPase activities, Cx43, and Kir2.1 while attenuating the surgical-induced increase in arrhythmia index and CK-MB and cTnI levels.	Pinocembrin alleviated ventricular arrhythmia in artery ligation-induced myocardial infarction in rats.
[50]	Ischemia and reperfusion cycle myocardial injury model in male Sprague–Dawley rats.	3, 10, or 30 mg/kg pinocembrin injected intravenously before ischemia.	Heart rate (HR) Mean arterial pressure (MAP) Rate-pressure product (RPP) Na^+^-K^+^-ATPase and Ca^2+^-Mg^2+^-ATPase activities Creatine kinase-MB isoenzyme (CK-MB) and cardiac troponin I (cTnI) levels. Gene and protein expression of Cx43, ZO-1, and Kir2.1.	Pinocembrin increases HR, MAP, RPP, Na^+^-K^+^-ATPase activities, Ca^2+^-Mg^2+^-ATPase activities, Cx43, ZO-1, and Kir2.1 while lowers the levels of CK-MB and cTnI.	Pinocembrin ameliorates ventricular arrhythmia in ischemia/reperfusion model in rats.

**Table 2 ijerph-17-03613-t002:** Clinical evidence on the cardioprotective effect of honey.

**Randomized Controlled Trial**					
**Ref**	**Study Population**	**Honey Treatment**	**Outcome Measures**	**Result**	**Conclusion**
[33]	Overweight or obese students from Mashhad University of Medical Science, Iran aged between 20 to 60 years old (*n* = 60).	70 g of oral natural unprocessed honey (Iran) dissolved in 250 mL tap water for a maximum of 30 days.	Body mass index (BMI) TG C-reactive protein (CRP) TC Low-density lipoprotein (LDL)	Treatment of honey reduced the BMI, serum TG, CRP, TC, and LDL after 30 days.	Consumption of honey reduces cardiovascular risk factors in overweight and obese adults.
[34]	Healthy male students of Isfahan University of Medical Sciences, Isfahan, Iran (*n* = 60).	70 g of oral natural honey (Iran) dissolved in 250 ml tap water for 6 weeks.	Total cholesterol (TC) Total triglyceride (TG) High-density lipoprotein (HDL)	Treatment of honey reduced TC but increased TG and HDL after 6 weeks.	Consumption of honey reduces cardiovascular risk factors in healthy male adults.
[35]	Postmenopausal women visiting outpatient clinics of Hospital Universiti Sains Malaysia (*n* = 100).	20 g oral Tualang honey (Malaysia) daily for 12 months.	Diastolic blood pressure (DBP) Systolic blood pressure (SBP) Lipid profile Fasting blood glucose	Treatment with honey reduced DBP and fasting blood glucose out of all the cardiovascular outcomes measured. SBP and lipid profile remain similar to the control.	Tualang honey supplementation reduces diastolic blood pressure and fasting blood glucose.
[42]	Children suffering from idiopathic dilated cardiomyopathy aged between 2 to 12 years old (*n* = 54).	1.2 g/kg oral Ziziphus honey (Yemen) daily for threeMonths, in addition to the standard heart failure medical therapy.	Ejection fraction (EF) Fraction shortening (FS)	EF and FS increased significantly in the honey group as compared with the control group	Honey consumption resulted in significant improvement in the EF and FS in a group of children suffering from IDCM.
**Non-Randomized Controlled Trial**					
**Ref**	**Study Population**	**Honey Treatment**	**Outcome Measures**	**Result**	**Conclusion**
[39]	Healthy and diabetic staff from the Dubai Specialized Medical Center and Medical Research Laboratories aged between 25 to 42 years old (*n* = 48)	90 g of oral natural honey (United Arab Emirates) dissolved in 250 mL drinking water for a maximum of 15 days.	Plasma glucose Insulin level CRP TG TC LDL HDL	Treatment of honey reduced level of plasma glucose, insulin, CRP, TC, LDL, and TG levels but caused elevation of HDL in healthy and diabetic subjects as early as 30 min, up to 15 days.	Consumption of honey reduces cardiovascular risk factors in healthy and diabetic adults.

**Table 3 ijerph-17-03613-t003:** Risk of bias assessment for pre-clinical reports on cardioprotective effect of honey.

		[17]	[19]	[20]	[21]	[22]	[26]	[27]	[28]	[29]	[33]	[34]	[35]	[37]	[38]	[40]	[41]	[42]	[43]	[44]	[45]	[46]	[47]
Selection	Numbers across groups were matched	+	+	+	+	+	+	+	+	+	+	+	+	+	+	+	+	+	+	+	+	+	
	Concealment of exposure allocation	+	+	+	+	+	+	+	+	+	+	+	+	+	+	+	+	+	+	+	+	+	
	Appropriate comparison group	+	+	+	+	+	+	+	+	+	+	+	+	+	+	+	+	+	+	+	+	+	
	Absence of confounding factors	+	+	+	+	+	+	+	+	+	+	+	+	+	+	+	+	+	+	+	+	+	
Performance	Identical experimental condition across groups	+	+	+	+	+	+	+	+	+	+	+	+	+	+	+	+	+	+	+	+	+	
	Blinded outcome assessor	-	-	-	-	-	-	-	-	-	-	+	+	+	+	+	+	+	+	+	+	+	
Attrition	Outcome data were complete with no exclusion from analysis	+	+	+	+	+	+	+	+	+	+	+	+	+	+	+	+	+	+	+	+	+	
Detection	Appropriate exposure assessment	+	+	+	+	+	+	+	+	+	+	+	+	+	+	+	+	+	+	+	+	+	
	Appropriate outcome assessment	+	+	+	+	+	+	+	+	+	+	+	+	+	+	+	+	+	+	+	+	+	
Reporting	All measured outcomes were reported	+	+	+	+	+	+	+	+	+	+	+	+	+	+	+	+	+	+	+	+	+	
Overall risk of bias		Low	Low	Low	Low	Low	Low	Low	Low	Low	Low	Low	Low	Low	Low	Low	Low	Low	Low	Low	Low	Low	Low

+ (Low risk of bias); - (high risk of bias).

**Table 4 ijerph-17-03613-t004:** Risk of bias assessment for clinical reports on the cardioprotective effect of honey.

		[30]	[31]	[32]	[36]	[39]
Selection	Random sequence generation	+	+	+	-	+
	Allocation concealment	+	+	+	-	-
Performance	Blinding of personnel	+	+	+	-	+
	Blinding of participants	+	+	+	-	-
Detection	Blinding outcome assessments	+	+	+	+	+
Attrition	Incomplete outcome data	+	+	+	+	+
Reporting	Selective reporting	+	+	+	+	+
Overall risk of bias		Low	Low	Low	Moderate	Low

+ (Low risk of bias); - (high risk of bias).

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
