# Peer review of "Cardioprotective Effects of Honey and Its Constituent: An Evidence-Based Review of Laboratory Studies and Clinical Trials"

_ijerph, 2020, doi:10.3390/ijerph17103613_

Round 1
Reviewer 1 Report
The review aims to evaluate the cardioprotective effects of honey. However, I have a few concerns as follows:
- The Cochrane Library database should be retrieved.
- In the Figure 1, the number of “Removal of duplicates” is vacancy.
- In the Table 1, the title “This is a table. Tables should be placed in the main text near to the first time they are cited” is incorrect. Additionally, it is inappropriate to analyze the animal studies and human studies together, they should be described respectively. For example, Table 1 can be used to describe the animal studies, Table 2 can be used to describe the human studies.
Author Response
Journal: IJERPH
Manuscript ID: ijerph-759281
Title of Paper: Cardioprotective Effects of Honey and its Constituent: A Comprehensive Review
Authors: Ruszymah Bt Hj Idrus, Nur Qisya Afifah Veronica Sainik, Abid Nordin, Aminuddin Bin Saim and Nadiah Sulaiman
We appreciate the time and efforts taken by the editor and reviewers in reviewing this manuscript. We have addressed all the issues indicated in the review report, and hopefully, the revised version can meet the journal publication requirements
Reviewer 1 comments:
The review aims to evaluate the cardioprotective effects of honey. However, I have a few concerns as follows:
- The Cochrane Library database should be retrieved.
Authors’ Response
Thank you for the insightful suggestion. The authors agree with the reviewers and have included Cochrane Library database in the search. Abdlrhman et al. (2017) study was retrieved from the aforementioned database.
- In the Figure 1, the number of “Removal of duplicates” is vacancy.
Authors’ Response
Thank you for pointing out this. The authors have conducted duplicate removal at the earlier stage of the screening process according to other reviewer’s suggestion. The number of duplicate records removed has been stated.
- In the Table 1, the title “This is a table. Tables should be placed in the main text near to the first time they are cited” is incorrect.
Authors’ Response
Thank you for pointing out this. The table titles were corrected to “Pre-clinical evidences on cardioprotective effect of honey” for Table 1 and “Clinical evidences on cardioprotective effect of honey” for Table 2.
- Additionally, it is inappropriate to analyze the animal studies and human studies together, they should be described respectively. For example, Table 1 can be used to describe the animal studies, Table 2 can be used to describe the human studies.
Authors’ Response
Thank you for the insightful suggestion. The authors agree with the reviewers and have separated the data of animal studies and human studies in the table and data synthesis.
Reviewer 2 Report
Minor revisions required:
Figure 1: Please include more details regarding the subject of the study in the title, as the authors provided a general title.
Table 1: No title was provided.
Table 1: Instead of using author-date style, please provide the number of the appropriate reference, as the Table is too long, and this may reduce its size.
Table 1: Attenuated Total Reflection–Fourier Transform Infra-Red may be replaced by its corresponding abbreviation (ATR-FTIR) as it is a well-known technique. The authors should include its significate as a footnote, or in the Abbreviations list.
Units: Please use SI (e.g., use “h”, “min”, and “mL” instead of “hours”, “minutes” and “ml” in Table 1).
Author Response
Journal: IJERPH
Manuscript ID: ijerph-759281
Title of Paper: Cardioprotective Effects of Honey and its Constituent: A Comprehensive Review
Authors: Ruszymah Bt Hj Idrus, Nur Qisya Afifah Veronica Sainik, Abid Nordin, Aminuddin Bin Saim and Nadiah Sulaiman
We appreciate the time and efforts taken by the editor and reviewers in reviewing this manuscript. We have addressed all the issues indicated in the review report, and hopefully the revised version can meet the journal publication requirements
Reviewer 2 comments:
Minor revisions required:
- Figure 1: Please include more details regarding the subject of the study in the title, as the authors provided a general title.
Authors’ Response
Thank you for the insightful suggestion. The authors have rephrased the figure title into “Flow chart of the selection procedure to identify relevant report on honey cardioprotective effect”.
- Table 1: No title was provided.
Authors’ Response
Thank you for pointing out this. The table titles were corrected to “Pre-clinical evidences on cardioprotective effect of honey” for Table 1 and “Clinical evidences on cardioprotective effect of honey” for Table 2.
- Table 1: Instead of using author-date style, please provide the number of the appropriate reference, as the Table is too long, and this may reduce its size.
Authors’ Response
Thank you for the insightful suggestion. The authors agree with the reviewers and have replaced the article ID into documentary style citation.
- Table 1: Attenuated Total Reflection–Fourier Transform Infra-Red may be replaced by its corresponding abbreviation (ATR-FTIR) as it is a well-known technique. The authors should include its significate as a footnote, or in the Abbreviations list.
Authors’ Response
Thank you for the insightful suggestion. The authors agree with the reviewers and have abbreviated ATR-FTIR in the Table 1. The authors also have included its significate in the Abbreviations list.
- Units: Please use SI (e.g., use “h”, “min”, and “mL” instead of “hours”, “minutes” and “ml” in Table 1).
Authors’ Response
Thank you for the insightful suggestion. The authors agree with the reviewers and have abbreviated SI units as suggested in the Table 1.
Reviewer 3 Report
This is an interesting review, however its methodology appears to mimic largely that of a systematic review, however there are a number of deviations. It may be more robust if done as a systematic review following a recognised approach e.g. PRISMA. So that quality of studies was assessed and potential sources of bias, as there could be potential for commercial interests with respect to honey especially with respect to some products e.g. manuka.
The flow chart for the study selection is confusing, in that it should be possible to identify duplicates at an earlier point.
The language in the abstract and to some extent in the rest of the paper could be scientifically more precise. The description of the pathology and mechanisms of cardiovascular disease could have more detail and physiological precision.
The decision to exclude cohort data, which was described as secondary would benefit from justification. As epidemiology is primary data, and it would be considered more valid to combine epidemiology and clinical trial data in humans together rather than combining human intervention studies with those from animals and in vitro models. Perhaps this paper would be better if it focused on either human or mechanistic studies. Currently the combined narrative is somewhat split and focusing on either of these would make for a much clearer and stronger paper.
Author Response
Journal: IJERPH
Manuscript ID: ijerph-759281
Title of Paper: Cardioprotective Effects of Honey and its Constituent: A Comprehensive Review
Authors: Ruszymah Bt Hj Idrus, Nur Qisya Afifah Veronica Sainik, Abid Nordin, Aminuddin Bin Saim and Nadiah Sulaiman
We appreciate the time and efforts taken by the editor and reviewers in reviewing this manuscript. We have addressed all the issues indicated in the review report, and hopefully, the revised version can meet the journal publication requirements
Reviewer 3 comments:
- This is an interesting review, however its methodology appears to mimic largely that of a systematic review, however there are a number of deviations. It may be more robust if done as a systematic review following a recognised approach e.g. PRISMA. So that quality of studies was assessed and potential sources of bias, as there could be potential for commercial interests with respect to honey especially with respect to some products e.g. manuka.
Authors’ Response
Thank you for pointing out this. Indeed, the authors abide to PRISMA approach in conducting the search. The authors have added risk of bias assessment accordingly in the Methods and Results section.
- The flow chart for the study selection is confusing, in that it should be possible to identify duplicates at an earlier point.
Authors’ Response
Thank you for pointing out this. The authors agree with the reviewers and have conducted the removal of duplicates at an earlier point for the search update.
- The language in the abstract and to some extent in the rest of the paper could be scientifically more precise. The description of the pathology and mechanisms of cardiovascular disease could have more detail and physiological precision.
Authors’ Response
Thank you for the insightful suggestion. The authors have added more detail and physiological precision on the pathology and mechanisms of cardiovascular disease in line 58-67.
- The decision to exclude cohort data, which was described as secondary would benefit from justification. As epidemiology is primary data, and it would be considered more valid to combine epidemiology and clinical trial data in humans together rather than combining human intervention studies with those from animals and in vitro models.
Authors’ Response
Thank you for pointing out this mistake. Indeed, epidemiology is a primary data and the list of secondary data was added in error. The authors did not exclude non-randomized epidemiology data. The human studies and animal studies have been separated accordingly.
- Perhaps this paper would be better if it focused on either human or mechanistic studies. Currently the combined narrative is somewhat split and focusing on either of these would make for a much clearer and stronger paper.
Authors’ Response
Thank you for the comments. The authors agree with the reviewer’s opinion. However, the authors aim to provide a comprehensive review of the cardioprotective effects of honey. Thus, the authors strongly believed that an inclusive approach on the two focuses fit the intended objective.
Reviewer 4 Report
The authors presented a study in which they highlight the effects of honey on cardiovascular diseases. The authors used three electronic databases (PubMed, Scopus, and MEDLINE) to identify reports on the cardioprotective effect of honey. Based on the pre-set eligibility criteria, qualified articles have been selected and discussed. Honey protects the heart via lipid metabolism improvement, antioxidative activity, blood pressure modulation, heartbeat restoration, myocardial infarct area reduction, antiaging properties, and cell apoptosis attenuation. The review established honey as a potential candidate to be explored further as the natural and dietary alternative to the management of cardiovascular diseases.
The study is interesting and the data are well presented. Therefore, the review deserves to be published in the International Journal of Environmental Research and Public Health, although I recommend the following changes before its publication (please, consult the attached file).

Author Response
Journal: IJERPH
Manuscript ID: ijerph-759281
Title of Paper: Cardioprotective Effects of Honey and its Constituent: A Comprehensive Review
Authors: Ruszymah Bt Hj Idrus, Nur Qisya Afifah Veronica Sainik, Abid Nordin, Aminuddin Bin Saim and Nadiah Sulaiman
We appreciate the time and efforts by the editor and reviewers in reviewing this manuscript. We have addressed all the issues indicated in the review report, and hopefully, the revised version can meet the journal publication requirements
Reviewer 4 comments:
The authors presented a study in which they highlight the effects of honey on cardiovascular diseases. The authors used three electronic databases (PubMed, Scopus, and MEDLINE) to identify reports on the cardioprotective effect of honey. Based on the pre-set eligibility criteria, qualified articles have been selected and discussed. Honey protects the heart via lipid metabolism improvement, antioxidative activity, blood pressure modulation, heartbeat restoration, myocardial infarct area reduction, antiaging properties, and cell apoptosis attenuation. The review established honey as a potential candidate to be explored further as the natural and dietary alternative to the management of cardiovascular diseases.
The study is interesting and the data are well presented. Therefore, the review deserves to be published in the International Journal of Environmental Research and Public Health, although I recommend the following changes before its publication (please, consult the attached file).
Thank you for the encouraging words. The authors were grateful for the reviewer's comments.
- The manuscript requires extensive editing of the English language/style.
Authors’ Response
Thank you for the comments. The authors have taken the necessary action as suggested.
- Considering the focus of the review: the cardioprotective effects of honey, I would suggest the authors extend section 1.3. Honey and its constituents, also adding ranges of values for both macro and micronutrients.
Authors’ Response
Thank you for the insightful suggestion. The authors agree with the reviewers and have expanded section 1.3 with honey constituents along with the averages of values of macro and micronutrients detected in honey in line 75-79.
- The authors argue that the origin of honey affects its bioactive potential, but few references have been made to this in the entire manuscript. I think it is crucial information, therefore I suggest the authors add bibliographic material on the matter.
Authors’ Response
Thank you for the insightful suggestion. The authors agree with the reviewers and have added empirical evidences of honey bioactive potential influenced by their origin as references [52-54] in line 362-364.
- Line 18. “..can lead to further complication”. Use the plural form “complications”.
Authors’ Response
Thank you for pointing out this. The edit was made accordingly in line 18.
- Line 50. “Hence, measurement..” Add “the” before “measurement”.
Authors’ Response
Thank you for pointing out this. The edit was made accordingly in line 50.
- Line 62. “..that can serve as a predictor to myocardial infarction”. Please change “to” with “of”.
Authors’ Response
Thank you for pointing out this. The edit was made accordingly in line 68.
- Lines 76-77. “..that honey positively affect risk factors..”. Please change with “..honey positively affects risk factors..”.
Authors’ Response
Thank you for pointing out this. The edit was made accordingly in line 88.
- Line 81. “..to identify and discusses all available..”. Please change with “..to identify and discuss all available..”.
Authors’ Response
Thank you for pointing out this. The edit was made accordingly in line 92.
- Line 93. “..that were published in English language..”. Please change with “..that were published in the English language..”.
Authors’ Response
Thank you for pointing out this. The edit was made accordingly in line 104.
- Line 119. “fulfil”, change with “fulfill”.
Authors’ Response
Thank you for pointing out this. The edit was made accordingly in line 140.
- Lines 125-127. Figure 1. The authors should add the number 26 at “Removal of duplicates”.
Authors’ Response
Thank you for pointing out this. The search has been updated. The number of duplicates removed was added accordingly.
- Lines 134-135. “Only researcher from Malaysia mentioned about the honey floral origin”. Please reformulate the sentence.
Authors’ Response
Thank you for pointing out this. The edit was made accordingly in line 154-155 to “Floral origin of honey was only mentioned in the three studies from Malaysia”.
- Lines 155-156. Table 1. Please, reformat.
Authors’ Response
Thank you for pointing out this. The table was reformatted and separated into 2 tables following the other reviewers suggestions.
- Line 165. “..for treatment of variety of disease..”. Please change with “..for treatment of a variety of diseases..”.
Authors’ Response
Thank you for pointing out this. The edit was made accordingly in line 217.
- Line 168. “In healthy rat..”. Please change with “In healthy rats..”.
Authors’ Response
Thank you for pointing out this. The edit was made accordingly in line 220.
- Line 170. “Honey also have..”. Please change with “Honey also has..”.
Authors’ Response
Thank you for pointing out this. The edit was made accordingly in line 222.
- Line 170. “..lipid metabolism in rat”. Please change with “..lipid metabolism in rats”.
Authors’ Response
Thank you for pointing out this. The edit was made accordingly in line 222.
- Line 176. “In human..”. Please change with “In humans”.
Authors’ Response
Thank you for pointing out this. Following suggestion from other reviewers, animal and human data synthesis were separated. The edit was made accordingly in line 185.
- Line 176. “..conducted a series of clinical study..”. Please change with “..conducted a series of clinical studies..
Authors’ Response
Thank you for pointing out this. The edit was made accordingly in line 185.
- Line 182. “They reported that honey caused reduction..”. Please change with “They reported that honey caused a reduction..”.
Authors’ Response
Thank you for pointing out this. The edit was made accordingly in line 191.
- Line 184. “..study done in young healthy patient..”. Please change with “..study done in young healthy patients..”.
Authors’ Response
Thank you for pointing out this. The edit was made accordingly in line 193.
- Lines 190-191. “..restores the activity of GRx and improve in the scavenging activities of SOD..”. Please change with “..restores the activity of GRx and improves the scavenging activities of SOD..”.
Authors’ Response
Thank you for pointing out this. The edit was made accordingly in line 234.
- Line 198. “When compared to the control, presence of honey was able..”. Please change with “When compared to the control, the presence of honey was able..”.
Authors’ Response
Thank you for pointing out this. The edit was made accordingly in line 241.
- Line 206. “In human..”. Please change with “In humans..”.
Authors’ Response
Thank you for pointing out this. Following suggestion from other reviewers, animal and human data synthesis were separated. Since Abd Wahab et al., (2018) study was grouped together with other human studies, the phrase “In human” was no longer used.
- Lines 206-207. “..with other bee product (95% honey, 4% bee bread, 1% royal jelly)..”. Please change with “..with other bee products (95% honey, 4% bee bread, 1% royal jelly)..”.
Authors’ Response
Thank you for pointing out this. The edit was made accordingly in line 196.
- Lines 212-213. “Development of the cardiac arrythmia model using ischemia and reperfusion cycle in isolated rat heart by Najafi et al. in 2011, enable studies on cardioprotective effect of honey to go beyond lipid profile and free radical”. The sentence is unclear, please reformulate it.
Authors’ Response
Thank you for the comment. The authors have reformulated the sentence into “Development of the ex vivo cardiac arrythmia model by Najafi et al. in 2011, enable studies on cardioprotective effect of honey to go beyond lipid profile and free radical. The model was established in isolated rat heart that was induced by ischemia and reperfusion cycle” in line 252-254. Hopefully, the edit will be clearer to present a paradigm shift of the cardiovascular parameters in ex vivo model of cardiac arrythmia.
- Line 261. “Treatment of chrysin show..”. Please change with “Treatment of chrysin shows..”.
Authors’ Response
Thank you for pointing out this. The edit was made accordingly in line 302.
- Line 263. “..that chrysin have..”. Please change with “..that chrysin has..”.
Authors’ Response
Thank you for pointing out this. The edit was made accordingly in line 304.
- Lines 269-270. “..utilized Nω-nitro-L-arginine methyl ester (L-NAME) to induce hypertension..”. Please change with “..utilized Nω-nitro-L-arginine methyl ester (L-NAME) to induces hypertension..”
Authors’ Response
Thank you for the comment. A verb follows by “to” do not have a singular form as it is a gerund. The current state of sentence is grammatically correct. Thus, the authors respectfully rebut this comment.
- Line 284. “..many of the drugs come with adverse effect”. Please change with “..many of the drugs come with adverse effects”.
Authors’ Response
Thank you for pointing out this. The edit was made accordingly in line 348.
- Line 285. “..has been an emerging trend among medical scientist”. Please change with “ ..has been an emerging trend among medical scientists”.
Authors’ Response
Thank you for pointing out this. The edit was made accordingly in line 349.
- Line 288. “..cardioprotective effect of Mediterranean diet..”. Please change with “..cardioprotective effect of the Mediterranean diet..”.
Authors’ Response
Thank you for pointing out this. The edit was made accordingly in line 351.
- Line 289. “Interest of using..”. Please change with “The interest of using..”.
Authors’ Response
Thank you for pointing out this. The edit was made accordingly in line 353.
- Line 294. “Many diseases have been associated to the imbalance..”. Please change with “Many diseases have been associated with the imbalance..”.
Authors’ Response
Thank you for pointing out this. The edit was made accordingly in line 358.
- Line 305. “With development of myocardial injury model in animal..”. Please change with “With development of myocardial injury model in animals..”.
Authors’ Response
Thank you for pointing out this. The edit was made accordingly in line 369.
- Lines 306-307. “Development of ischemia and reperfusion cycle in isolated rat heart, has enable the measurement of cardiac arrythmia and myocardial infarct area”. The sentence is unclear, please reformulate it.
Authors’ Response
Thank you for the comment. The authors have reformulated the sentence into “For example, cardiac arrythmia and myocardial infarct size can be measured from the ex vivo model created by subjecting isolated rat heart to ischemia and reperfusion cycle” in line 370-372. Hopefully, the edit will be clearer to present expansion of cardiac parameters to be measured following development of cardiac injury model.
- Line 312. “..of honey against aged-related pathology..”. Please change with “..of honey against agerelated pathology..”.
Authors’ Response
Thank you for pointing out this. The authors believed that reviewer suggested removal of the hyphen only, not the space between these two words entirely. The edit was made accordingly in line 377.
- Lines 316-320. Please, rewrite the conclusions. Here a suggestion: The review has successfully identified relevant evidence on the honey cardioprotective effect. Honey is able to: modulate oxidation, reduce blood pressure, restore heartbeats, reduce myocardial infarct area, improve lipid metabolism, exert antiaging properties, and attenuate cell apoptosis. Honey demonstrated to be a potential candidate as a natural alternative for the management of cardiovascular disease.
Authors’ Response
Thank you for the insightful suggestion. The authors agree with the reviewer’s suggestion but not on the use of colon within the paragraph. Hence, the conclusion was edited into “The review has successfully identified relevant evidence on the honey cardioprotective effect. Honey is able to modulate oxidation, reduce blood pressure, restore heartbeats, reduce myocardial infarct area, improve lipid metabolism, exert antiaging properties, and attenuate cell apoptosis. Honey demonstrated to be a potential candidate as a natural alternative for the management of cardiovascular disease.” in line 381-385.
- Replace the word "arrhythmia" with "arrhythmia" throughout the manuscript.
Authors’ Response
Thank you for pointing out this. The edit was made accordingly within the entire manuscript.
Round 2
Reviewer 3 Report
It is appreciated how the authors have revised the manuscript.
The view that the paper should include both animal data and human data is logically argued. However, the lack of epidemiological data, means it is not comprehensive. There are two solutions to this issue, rename the paper and revise the methodology to say it is a comprehensive review of laboratory studies and clinical trials. Or, re-run the search as there are a number of epidemiological studies reporting cardiovascular effects of honey.
Author Response
Journal: IJERPH
Manuscript ID: ijerph-759281
Title of Paper: Cardioprotective Effects of Honey and its Constituent: A Comprehensive Review
Authors: Ruszymah Bt Hj Idrus, Nur Qisya Afifah Veronica Sainik, Abid Nordin, Aminuddin Bin Saim and Nadiah Sulaiman
We appreciate the time and efforts taken by the editor and reviewers in reviewing this manuscript. We have addressed all the issues indicated in the review report, and hopefully, the revised version can meet the journal publication requirements
It is appreciated how the authors have revised the manuscript.
Thank you for the kind and encouraging words.
The view that the paper should include both animal data and human data is logically argued. However, the lack of epidemiological data, means it is not comprehensive. There are two solutions to this issue, rename the paper and revise the methodology to say it is a comprehensive review of laboratory studies and clinical trials. Or, re-run the search as there are a number of epidemiological studies reporting cardiovascular effects of honey.
The authors accept the reviewer's argument and renamed the title into "Cardioprotective Effects of Honey and its Constituent: An Evidence-Based Review of Laboratory Studies and Clinical Trials". In the method, the word "clinical studies" was changed to "clinical trials" to reflect this amendment.